# If Smoking Were Eliminated, Which US Counties Would Still Have High Rates of Smoking-Related Cancers?

**DOI:** 10.3390/ijerph192215292

**Published:** 2022-11-19

**Authors:** Douglas J. Myers, David Kriebel

**Affiliations:** 1School of Public and Population Health, College of Health Sciences, Boise State University, Boise, ID 83725, USA; 2Lowell Center for Sustainable Production, University of Massachusetts Lowell, Lowell, MA 01854, USA

**Keywords:** smoking-related cancers, environmental pollution, county level analysis, environmental quality index

## Abstract

Objective: to characterize the county variability of the impact of smoking elimination on rates of smoking-related cancers and explore whether common environmental indices predicted which metropolitan counties would experience high rates of smoking-related cancers even after smoking was eliminated. Methods: Surveillance, Epidemiology, and End Results Program (SEER) and Environmental Protection Agency (EPA) data were obtained. County level cancer rates for 257 metropolitan SEER counties, including the observed rates and those predicted after eliminating smoking, were derived via multilevel regression modeling and age standardized to the 2016 SEER population. Associations between the EPA’s Environmental Quality Index (EQI) scores and “Low Benefit” counties (counties that remain above the top 20th percentile of post-smoking elimination incidence rates) were explored via logistic regression. Results: Reductions in smoking-related cancer incidence ranged from 58.4 to 3.2%. The overall EQI (OR = 1.96, 95% CI [1.34, 2.86]) and the air quality index (OR = 5.99, 95% CI [3.20, 11.22]) scores predicted higher odds of being a “Low Benefit” county. Conclusions: Substantial inequities in the post-smoking elimination cancer rates were observed; air pollution appears to be a primary explanation for this. Cancer prevention in metropolitan counties with high levels of air pollution should prioritize pollution control at least as much as tobacco control.

## 1. Introduction

Smoking is the single most important modifiable cause of cancer [1], and the substantial declines in smoking rates in the U.S. over the past few decades have resulted in impressive reductions in the incidence of cancers of the lungs and other smoking-related types [2]. In a previous paper [3], we confirmed these findings using multilevel regression methods to estimate the association between county level smoking prevalence and cancer incidence data for 612 of the approximately 3100 counties in the U.S., which provided data for the Surveillance, Epidemiology, and End Results (SEER) program for 12 types of cancer that are known to be caused by tobacco.

We found that, if smoking was entirely eliminated, the incidence of these 12 types of cancer would decrease by about 40%, which would translate to a 16.3% reduction in all types of cancer combined [3]. This finding is in good agreement with other authors using different methods [1]. Our previous paper went beyond confirming this substantial role of tobacco in cancer, however. The county data and multilevel regression methods allowed us to estimate the county-by-county variability in tobacco’s contribution, and to observe that not all counties would benefit equally from smoking elimination [3].

Some counties would see only modest reductions in the incidence of smoking-related cancers, according to this modeling approach. These “low benefit” counties tended to be in metropolitan areas, which suggested that perhaps carcinogenic exposures such as air pollution or occupational agents might be “offsetting” the potential benefit that would be expected if smoking could be eliminated. The five counties with the highest predicted cancer rates in 2016 after eliminating smoking were all in the metropolitan areas of large cities: Jefferson County KY (Louisville); Wayne and Macomb counties MI (Detroit); Campbell County KY (Cincinnati); and Jefferson Parish LA (New Orleans). If smoking was eliminated, we predicted that these five counties would see only an approximate 8% reduction in their rates of smoking-related cancers—far less than the overall average of about 40% after total smoking elimination.

In the current paper, we investigated the question of why some counties would retain high rates of smoking-related cancers if smoking was eliminated. Specifically, we tested hypotheses that these low-benefit counties would have higher exposures to environmental carcinogens. Likely candidates included PM_2.5_ and other carcinogenic components of urban air pollution.

## 2. Materials and Methods

### 2.1. Cancer Data

Cancer incidence data for 2016 were obtained from the Surveillance, Epidemiology, and End Results (SEER) program of the National Cancer Institute (NCI) [4]. The program includes 18 cancer registries from across the United States. Sixteen registries were included in this analysis (Alaska and Hawaii were excluded); these 16 registries cover approximately 20% of the 2016 US population. SEER data contain cancer incidence information, as well as patient demographics. Population data are also provided by the SEER program. As the SEER incidence data provide county of residence information, the county was the unit of analysis (as it was in our previous analysis). Because our previous study found that the “low benefit” counties were all in metropolitan areas, the present analyses were restricted to the 257 SEER counties classified as metropolitan by the US Department of Agriculture (Rural and Urban County Codes 1, 2, and 3) [5]. These constitute 42% of the 612 counties in the previous SEER analyses. Counties identified as metropolitan, regardless of population size, were grouped into a single “metropolitan” category.

In line with our previous study [3], we chose to study the 12 cancer types which are deemed to be caused by smoking according to the U.S. Centers for Disease Control and Prevention [6]. These 12 are:
Trachea, bronchus, and lung (ICD-O-3 codes C33.9–34.9);Larynx (C32.0–32.9);Oral cavity and pharyngeal (C00–14.8);Esophagus (C15.0–15.9);Stomach (C16.0–16.9);Colon and rectum (C18.0–20.9);Liver (C22.0);Pancreas (C25.0–25.9);Kidney and renal pelvis (C64.9–65.9);Urinary bladder (C67.0–67.9);Cervix (C53.0–53.9);Acute myeloid leukemia (ICD-O-3 histology codes 9840, 9861, 9865–9867, 9869, 9871–9874, 9895–9898, 9910–9911, and 9920).

While these cancers are associated with smoking, not all incidence of these cancer types is caused by smoking. Previous research indicated [3] that the county level incidence rates of these types of cancer would drop by approximately 40%, on average, if smoking had been eliminated twenty years prior.

### 2.2. Smoking, Environmental, and Demographic Data

Independent variables were available at either the individual level or county level. Individual level variables for each case include sex and age, examined in 5-year categories (20–24, …, 80–84). As in the original analysis, race effects were not modeled (and all races combined were included in the dataset) because of the small numbers of non-whites in many of the SEER counties.

Reliable individual smoking data are not available from SEER. Therefore, county level smoking prevalence estimates were used instead. We used the age-standardized calendar year- and sex-specific smoking prevalence obtained from the Institute for Health Metrics and Evaluation [7]. These smoking prevalence estimates were based on Behavioral Risk Factor Surveillance System (BRFSS) data, which were modeled to generate estimates of county level smoking prevalence for the US between 1996 and 2012 for ages 18 and over. Estimates for counties that had limited data for a given year were derived via spatial and temporal smoothing techniques which included county and state-level covariates. The smoking variable was defined as “prevalence of current daily cigarette smoking”. Smoking prevalence estimates for 1996 were used as they allowed for a lag of 20 years for the analysis of the 2016 SEER cancer incidence data.

We hypothesized that environmental exposures might explain why some counties would expect low benefits in cancer reduction from smoking elimination. Candidate carcinogenic exposures included PM_2.5_ as well as other pollutants in air, water, and land. County level PM_2.5_ data were gathered from the Center for Air, Climate, and Energy Solutions (CACES). These estimates were derived using spatially decomposed v1 empirical models, as described in [8]. The earliest available year for these data was 1999; therefore, this variable was lagged 17 years when used to model the 2016 SEER incidence data. For other environmental exposures, we used the U.S. Environmental Protection Agency’s Environmental Quality Index (EQI) [9]. The EQI was designed to measure overall environmental quality at the county level, with the goal of improving the current understanding of the relationship between environmental conditions and human health. This objective seemed to fit well with our goal of investigating the role of environmental exposures in explaining inequalities in the benefit of smoking elimination.

The EQI is composed of indices representing five environmental domains: air, water, land, built environment, and sociodemographic. The earliest available data represent the period 2000–2005, and these were used for the primary analyses to maximize the latency for the 2016 cancer incidence data. The most recent data, 2006–2010, were used in a sensitivity analysis investigating the impact of the choice of latency on the results.

Each of the five environmental quality scales was constructed by combining information from a large number of environmental exposure variables using principal component analysis [10]. The five resulting scales or sub-indices are unitless numbers which can be considered county rankings of environmental quality. The overall EQI is a combined score including all five domains. We hypothesized a priori that the air quality index (AQI) would be the most important of the five in explaining variations in cancer rates because of the well-established contribution of air pollutants to cancer risk [11,12]. The AQI combines data on particulates (both PM_2.5_ and PM_10_) and the other EPA criteria air pollutants including nitrogen dioxide, sulfur dioxide ozone, carbon monoxide, as well as hazardous air pollutants (HAPs) including a number of carcinogens such as 1,2-dibromo-3-chloropropane, benzidine, carbon tetrachloride, chloroprene, ethylene dibromide, formaldehyde, trichloroethylene, and vinyl chloride [10].

The other four indices are water, representing overall water quality and chemical contamination of surface and drinking water; land, which includes measures of land use, pesticide use, presence of industrial facilities, and radon exposure; sociodemographic, which represents socioeconomic factors including poverty and crime; and built environment, which includes data on pedestrian and highway safety, access to various businesses including food, healthcare, and recreation and the quality of housing stock.

### 2.3. Modeling Actual and Simulated Cancer Rates

Statistical analyses were performed using multilevel mixed-effects regression models in STATA/MP 16.0 [13]. We modeled cancer incidence rates by using observed cancer counts as the dependent variable and population as the offset. The observed counts were modeled assuming a negative binomial distribution, based on the presence of over-dispersion; incidence rate ratios and 95% confidence intervals were generated. The fixed-effect part of the model included age, sex, and county level age-adjusted and sex-stratified daily smoking prevalence, lagged 20 years. Smoking prevalence was modeled as a fixed effect because we assumed the effect of smoking prevalence would be constant across counties. County specific intercepts were included in the random component. Only metropolitan counties were included in the model.

After fitting the model, we used STATA’s “predict” command to generate two predicted counts of cancers: one derived using the actual smoking prevalence values, the second assuming smoking was completely eliminated (daily smoking prevalence = 0). Values were then converted to the expected county rates using county population data. The fixed- and random-effects model components were used to create the predicted values. To allow for comparison across counties, each county’s rate was age standardized to the sex-specific SEER population distribution for 2016.

### 2.4. Analyzing Environmental Predictors of Low-Benefit Counties

Counties which the model predicted would still have high rates of the 12 types of smoking-related cancers, even after smoking was eliminated, were labelled “low benefit” counties (Figure 1). Specifically, this term was assigned to those SEER metropolitan counties in the top 20% of the distribution of cancer incidence rates after eliminating smoking (the remaining counties were referred to as “high benefit” counties). This 20% cut point corresponded to an incidence rate in 2016 greater than 235/100,000 after smoking was eliminated (this cut point represented 19.8% of the metropolitan SEER counties, and those counties included 22.0% of the entire SEER population in 2016).

T-tests were used to evaluate differences in demographic variables, PM_2.5_, and EQI scores by “low benefit” status. Logistic regression was used to examine associations between being a low-benefit county and environmental quality indices. Bivariate and multivariate analyses were performed. Akaike’s information criteria (AIC) were used to compare the model fit.

Finally, to determine whether differences in environmental variables by low-benefit status was due to residual confounding by county level smoking status, we conducted a sensitivity analysis to examine differences among counties near the mean of smoking prevalence (within +/−0.67 standard deviations of the mean). This reduced the variance in county level smoking prevalence and allowed us to compare results produced using all metropolitan counties to this subset with similar smoking prevalence rates (Appendix A).

## 3. Results

There were 136,158 cases of smoking-related cancers identified in the 257 metropolitan SEER counties in 2016. The population in these counties was 58,129,876, which represents 90.9% of the population in the full set of 607 counties included in the 16 SEER registries.

The base model used to generate the 2016 county cancer incidence rates showed the anticipated strong associations between cancer incidence and age, gender, and smoking prevalence (lagged 20 years) (Table 1). Males had higher rates of smoking-related cancers than females (RR = 1.42, 95% CI 1.39–1.46), and the rates rose steadily across the five-year age groups. Likewise, the rate of cancers increased steadily across the categories of smoking prevalence, and the well-described strong contribution of smoking to the rates of these 12 cancer types is clearly evident.

### 3.1. Descriptive Statistics for All Metropolitan Counties

After fitting a model including age, gender, and smoking prevalence (similar to the model in Table 1 except that the smoking prevalence lagged 20 years was coded as a continuous value rather than in six levels), the average predicted cancer incidence rate for all 257 metropolitan counties was 275.1 per 100,000 (Table 2). After setting all county smoking prevalences to zero, the model predicted an average incidence rate of 201.9, an average reduction of 25.0%. The model predicted considerable variability from county to county in the “post-smoking elimination” cancer rates, however, and the percent reduction in cancer incidence rates after smoking elimination ranged as high as 58.4% to a low of 3.2%; post-smoking elimination incidence rates ranged from 137.0/100,000 to 292.0/100,000 (Figure 1).

### 3.2. Comparing Low-Benefit and High-Benefit Counties

After simulating the complete elimination of smoking, some counties were predicted to benefit only modestly (Figure 1 and Table 2). The one-fifth of metropolitan counties (n = 51) with the highest predicted cancer rates post-smoking elimination (the “low benefit” counties) would have cancer incidence rates only eight percent lower than the actual mean rate for all SEER metropolitan counties in 2016. What distinguished these counties from other metropolitan counties?

The concentrations of PM_2.5_, lagged 17 years, did not distinguish low- from high-benefit counties (*p* = 0.32), nor was there an important difference in county population size, which might have been a proxy for urban environmental hazards. However, the overall EQI index was more than twice as high (poorer environmental quality) in the low-benefit counties (0.98 versus 0.45, *p* < 0.001).

The five sub-indices that comprise the EQI were studied separately for differences between low- and high-benefit counties and the Air Quality Index showed the most striking difference (1.24 vs. 0.62, *p*-value < 0.001), indicating substantially poorer air quality in the low-benefit counties (Table 2). The only other notable difference was in the Built Environment Index (0.66 vs. 0.01, *p*-Value < 0.01), in which the low-benefit counties actually had a higher, meaning better, score.

Because the regression model including county smoking prevalence was used to predict cancer rates under the hypothetical condition that smoking had been eliminated (essentially “controlling for” effect of smoking), one might expect to find no difference in smoking prevalence between low- and high-benefit counties. In fact, the low-benefit counties had modestly higher 20-year lagged smoking prevalence (23.1% vs. 21.5%, *p* = 0.03). This modest residual difference in smoking prevalence between low- and high-benefit counties probably represents confounding effects of other factors that were correlated with county smoking prevalence.

Bivariate logistic regression modeling found that both EQI and air quality index scores predicted higher odds of being a low-benefit county (Part a of Table 3). A one standard deviation increase in the EQI was associated with nearly double the odds of being a low-benefit county (OR = 1.96, 95% CI [1.34, 2.86]); a one unit increase in the air quality index indicated a six-fold increase in the odds of being a low-benefit county (OR = 5.99, 95% CI [3.20, 11.22]). The Built Environment Quality Index was also associated with increased odds of being a low-benefit county (OR = 2.70, 95% CI [1.68, 4.32]); however, of these two sub-indices, AQI produced the better fitting model (AIC = 219.4 versus 237.2). When the five sub-indices were included together in a single model, only the air quality index remained an important predictor of low-benefit counties (adjusted OR = 4.43, 95% CI [2.14, 9.19]) (Part b of Table 3).

Two additional analyses were conducted to evaluate potential limitations in the available data. First, it is well known that the epidemiologic assessment of environmental carcinogens should take into account long latencies between exposure and disease. The EQI datasets cover two time periods, either 2000–2005, which was used above, or 2006–2010. Thus, the maximum available latency was 12 to 17 years. Using these data, the odds ratio associating the AQI with being a low-benefit county in 2016 was 5.99 (Table 3a). When the more recent AQI score was used, this odds ratio decreased substantially to 1.90 (results not shown). This reduction may suggest that the longer latency was more appropriate, as expected. However, caution is warranted in this interpretation because the Environmental Protection Agency changed the makeup of the EQI measure between the two time periods, making direct comparisons difficult.

The second sensitivity analysis was conducted to investigate further the residual difference in smoking prevalence between low- and high-benefit counties (Table 2). We were concerned that this residual difference might be a proxy for important unmeasured exposures. We therefore repeated the analyses in Table 2 using a restricted dataset of only the metropolitan counties whose smoking prevalence was very close to the mean for all metropolitan counties. Specifically, we limited the dataset to the 133 metropolitan counties with smoking prevalence within 0.67 standard deviations of the mean (Appendix A). As in the full analysis (Table 2), the AQI remained higher (poorer air quality) in the low-benefit counties, and the magnitude of the difference with high-benefit counties actually increased slightly. Thus, we conclude that the findings in Table 2 and Table 3 on the importance of air quality were not due to residual confounding by county smoking prevalence.

## 4. Discussion

These analyses suggest that metropolitan counties with poorer air quality would benefit less from tobacco control, and retain higher smoking-related cancer incidence rates, than counties with cleaner air. There are, however, several limitations. We lacked individual smoking data and were constrained to using county averages. Smoking data indicated county prevalence; no measure of smoking intensity or duration was available. However, as noted earlier, despite this limitation, the resulting estimate of the mean contribution of smoking to the incidence of the smoking-related cancers was in good agreement with other authors [1,3]. Another limitation was that the EQI data were not available with a 20-year latency, which would have been preferable.

There are both strengths and limitations to the AQI being an integrated measure of many different pollutants, considering not only particulate matter but also many organic and inorganic pollutants including carcinogens. From a policy perspective, it may be useful to have an overall measure of air pollution, albeit with the disadvantage that the contributions of individual pollutants are not identifiable. We were able to separately investigate the role of PM_2.5_, and the finding that the AQI was more strongly associated with being a low-benefit county than PM_2.5_ points to the likelihood that there are other important carcinogens in urban air besides particulate matter. In future work, we will use more refined measures of air toxics to pursue this hypothesis.

The finding that counties with poorer air quality would benefit less from tobacco control could be understood as simply another way to say that air pollution causes some of the same types of cancers as tobacco, which is now well accepted [11,12]. For example, very recent work by Swanton and colleagues has identified a potentially powerful mechanism through which PM_2.5_ can contribute to non-small cell lung cancer (NSCLC) in non-smokers, a disease with a high frequency of EGFR mutations (EGFRm). The authors found that PM promotes precursor lung epithelial cells initiated by EGFRm [14].

What remains controversial is the question of the relative importance of environmental exposures versus “lifestyle” factors in cancer prevention [3,15,16,17]. By simulating the pattern of county cancer rates in a world without tobacco, we find that there would be substantial inequities in the remaining cancer rates, and that urban air pollution, in all its considerable complexity, appears to be a primary explanation for this.

## 5. Conclusions

Often, when the priorities for cancer prevention are discussed, the overall or average contributions of tobacco, diet, occupational exposures, air pollution, etc., are compared, with the conclusion that smoking is by far the most important. While this argument appears logical, it overlooks the fact that there will be significant inequities in these contributions depending on environmental conditions. More specifically, we conclude that cancer prevention in metropolitan counties with high levels of air pollution should prioritize pollution control at least as much as tobacco control.

## Figures and Tables

**Figure 1 ijerph-19-15292-f001:**
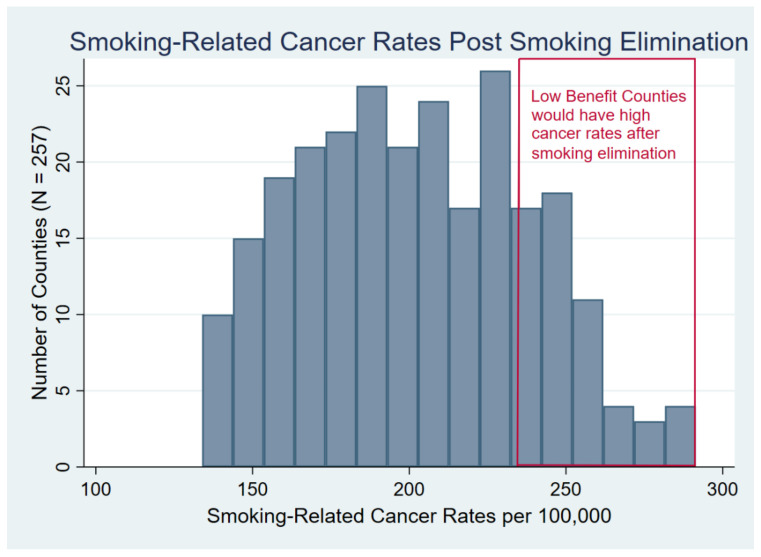
Distribution of county-level incidence rates of 12 smoking-related cancers after smoking was eliminated. Counties with rates in the top 20th percentile (greater than 235/100,000) were identified as “Low Benefit”.

**Table 1 ijerph-19-15292-t001:** Incidence rate ratios for determinants ^a^ of 12 smoking-related types of cancer ^b^ in the 257 SEER metropolitan counties, 2016—results of a multilevel negative binomial regression model.

	IRR ^†^	95% CI ^‡^	
Age Group			
20–24	1	(Ref.)	
25–29	1.85	1.62	2.10
30–34	3.42	3.03	3.85
35–39	5.78	5.15	6.48
40–44	9.28	8.30	10.38
45–49	15.82	14.18	17.64
50–54	30.69	27.57	34.17
55–59	46.94	42.19	52.22
60–64	67.21	60.43	74.75
65–69	93.44	84.02	103.90
70–74	124.34	111.81	138.28
75–79	152.63	137.22	169.77
80–84	173.00	155.48	192.50
Sex			
Female	1	(Ref.)	
Male	1.42	1.39	1.46
Smoking Prevalence			
4.0%<10.1%	1	(Ref.)	
10.1%<12.2%	1.14	1.02	1.28
12.2%<15.7%	1.21	1.09	1.34
15.7%<19.2%	1.28	1.15	1.43
19.2%<22.9%	1.39	1.24	1.56
22.9%<38.3%	1.66	1.47	1.88

^†^ Incidence rate ratio 95% confidence interval; ^‡^ 95% confidence interval; ^a^ smoking-related cancer sites classified according to [6]. ^b^ Estimates from a negative binomial regression model with random intercept on county and random slope on year. SEER 16 registries, 2016, Alaska Native Tumor Registry excluded because of the lack of information on county. Also excludes Hawaii.

**Table 2 ijerph-19-15292-t002:** Incidence rates of 12 smoking-related types of cancer in the 257 SEER metropolitan counties, 2016, as observed and assuming smoking elimination, and covariates.

Mean Cancer Incidence and Covariates	All Metropolitan Counties	Counties with Post-Smoking Elimination Incidence Rates above and below the 20th Percentile *		
		Above	Below	*t*-Test
		(Low benefit)	(High Benefit)	
	(n = 257)	(n = 51)	(n = 206)	*p*-Value
Cancer Rate per 100,000—Observed	275.1	301.5	268.6	<0.01
Cancer Rate—Smoking Eliminated	201.9	252.6	189.3	<0.01
Cancer Rate—Percent Reduction	25.0%	15.6%	27.3%	<0.01
County Population	226,186	250,342	220,206	0.73
PM_2.5_ (μg/m^3^)	14.1	13.7	14.3	0.32
Environmental Quality Index	0.56	0.98	0.45	<0.01
Air Quality Index	0.74	1.24	0.62	<0.01
Water Quality Index	0.08	−0.05	0.12	0.28
Land Quality Index	0.18	0.26	0.16	0.46
Built Environment Quality Index **	0.13	0.66	0.01	<0.01
Sociodemographic Quality Index **	0.47	0.65	0.43	0.17
Smoking Prevalence (%, lagged 20 years)	21.8	23.1%	21.5%	0.03

Data are for 257 SEER metropolitan counties classified by their predicted smoking-related cancer rates after smoking elimination. * Top 20th percentile: Incidence rate >235/100,000. ** Valences of the Built Environment and Sociodemographic Indices are opposite the overall EQI, Air, Water, and Land Indices, and higher values indicate better environmental conditions.

**Table 3 ijerph-19-15292-t003:** (**a**) Association between environmental exposures and low-benefit county status for 257 SEER metropolitan counties, 2016—results of bivariate logistic regression models. (**b**) Association between environmental exposures and low-benefit county status for 257 SEER metropolitan counties, 2016—conditional results from a multivariate logistic regression model.

(a)				
	Crude			
Exposures	OR ^†^	95% CI ^‡^		AIC ^§^
Environmental Quality Index	1.96	1.34	2.86	246.2
Air Quality Index	5.99	3.20	11.22	219.4
Water Quality Index	0.85	0.63	1.15	259.0
Land Quality Index	1.15	0.79	1.67	259.5
Built Environment Quality Index *	2.70	1.68	4.32	237.2
Sociodemographic Quality Index *	1.23	0.91	1.64	258.3
PM_2.5_ (μg/m^3^)	0.96	0.88	1.04	259.1
Smoking Prevalence	1.08	1.01	1.15	255.4
**(b)**				
	**Adjusted**			
**EPA Exposure Indices**	**OR ^†^**	**95% CI ^‡^**		AIC ^§^
Air Quality Index	4.43	2.14	9.19	223.4
Water Quality Index	0.80	0.56	1.14	
Land Quality Index	1.00	0.62	1.61	
Built Environment Quality Index *	1.59	0.90	2.82	
Sociodemographic Quality Index *	0.96	0.67	1.37	

* Valences of the Built Environment and Sociodemographic Indices are opposite the overall EQI, Air, Water, and Land Indices, and higher values indicate better environmental conditions. ^†^ Odds ratio. ^‡^ 95% confidence interval; **^§^** Akaike’s information criteria.

## Data Availability

Data were obtained from the US Government’s Surveillance Epidemiology and End Results (SEER) Program. https://seer.cancer.gov/csr/1975_2016/ (accessed on 11 June 2019).

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
