# Peer review of "If Smoking Were Eliminated, Which US Counties Would Still Have High Rates of Smoking-Related Cancers?"

_ijerph, 2022, doi:10.3390/ijerph192215292_

Round 1
Reviewer 1 Report
The paper provides a novel approach to address a timely question that deserves more attention: the relationship between air quality and cancer risk. The report was well written, and the methods were described in considerable detail and well explained. The authors provide provocative findings that could stimulate additional discussion and debate. The following comments are offered for the authors’ consideration:
1. The authors state that CDC had deemed 12 cancer types as caused by smoking (line 74-75). In citation #6, CDC acknowledged that cases of tobacco-associated cancer “might or might not be in persons who used tobacco.” The authors are encouraged to clarify that cancers labelled by CDC as “tobacco-associated” are not necessarily caused by smoking; in fact, many such cancers are also diagnosed among people who do not smoke or have never smoked. This is true even for lung cancer. A 2021 study reported that 12.5% of lung cancer cases in 7 states occurred among persons who had never smoked (Siegel D, et al, JAMA Oncol. 2021;7(2):302-304. https://doi.org/10.1001/jamaoncol.2020.6362).
2. Citation #2 is the Annual Report to the Nation from 2017; this report is updated annually and the report for 2022 has just been published (Cronin, KA, Scott, S, Firth, AU, et al. Annual report to the nation on the status of cancer, part 1: National cancer statistics. Cancer. 2022; 1- 34. https://doi.org/10.1002/cncr.34479). That report indicated between 2014-2018, the incidence of lung cancer declined among males (AAPC=-2.6%) and females (AAPC=-1.1%), and the authors attributed these declines to “the continuous decline in smoking prevalence” (p.28). However, during this same period, the incidence of several “smoking-related cancers” increased among males and/or females: pancreas, kidney and renal pelvis, liver, oral cavity and pharynx, and cervix. Arguably, factors unrelated to smoking are driving the increasing trends in these cancers. Combining these cancers together with lung cancer could distort estimates of the impact of changes in smoking prevalence on cancer rates.
3. The authors are encouraged to revise their tables and figure to be “stand alone” without reference to the text as much as possible. Titles should be descriptive, with details provided in footnotes. In Table 2, it appeared that county population was misplaced in the first column. Careful proof-reading may be needed as the paper is processed by this journal to avoid errors.
4. Had the authors considered limiting their analyses to lung cancer, or to conducting a sensitivity analysis that examined specifically lung cancer? Or perhaps this could be considered as a potential area for future research.
5. Because smoking prevalence is a key variable in the analysis, the authors are encouraged to provide further details about their methods for generating estimates of county-level smoking prevalence from BRFSS data. Were there counties for which BRFSS data were not available? If so, how did the methods used to generate county estimates compare with the those used by Berkowitz et al. in 2016 (Cancer Epidemiol Biomarkers Prev (2016) 25 (10): 1402–1410. https://doi.org/10.1158/1055-9965.EPI-16-0244 )?
6. In the discussion, the authors state that it is well accepted that air pollution causes some of the same types of cancer as tobacco (line 292-293). The cited references support a causal association between outdoor air pollution and lung cancer but not between outdoor air pollution and the 11 other “smoking-related” cancers. The authors are encouraged to include additional citations, or identify areas for future research, related to air pollution and cancers other than lung cancer.
7. A presentation by Charles Swanton at European Society for Medical Oncology (ESMO) Congress 2022 in Paris generated a great deal of recent media attention because it proposed a novel biologic mechanism by which air pollution could synergistically contribute to cancer development. For example, see this ASCO Post: https://ascopost.com/issues/october-10-2022/mechanism-linking-air-pollution-to-lung-cancer-identified/. This work might be relevant to the authors’ discussion of their results and the potential importance of pollution control. A preprint of that study: Swanton, Charles, William Hill, Emilia Lim, Claudia Lee, Clare Weeden, Marcellus Augustine, Kezhong Chen et al. "Non-Small-Cell Lung Cancer Promotion by Air Pollutants." (2022) in available online at: https://assets.researchsquare.com/files/rs-1770054/v1/05c2bee9-d3dc-4cfc-b19c-3e488ff0006a.pdf?c=1663076245
Author Response
Comments and Suggestions for Authors
The paper provides a novel approach to address a timely question that deserves more attention: the relationship between air quality and cancer risk. The report was well written, and the methods were described in considerable detail and well explained. The authors provide provocative findings that could stimulate additional discussion and debate. The following comments are offered for the authors’ consideration:
- The authors state that CDC had deemed 12 cancer types as caused by smoking (line 74-75). In citation #6, CDC acknowledged that cases of tobacco-associated cancer “might or might not be in persons who used tobacco.” The authors are encouraged to clarify that cancers labelled by CDC as “tobacco-associated” are not necessarily caused by smoking; in fact, many such cancers are also diagnosed among people who do not smoke or have never smoked. This is true even for lung cancer. A 2021 study reported that 12.5% of lung cancer cases in 7 states occurred among persons who had never smoked (Siegel D, et al, JAMA Oncol. 2021;7(2):302-304. https://doi.org/10.1001/jamaoncol.2020.6362).
Response: We added to lines 90-93: While these cancers are associated with smoking, not all incidence of these cancer types are caused by smoking. Previous research indicated that county incidence of these types of cancer would drop by approximately 40%, on average, if smoking had been eliminated twenty years prior.
2.Citation #2 is the Annual Report to the Nation from 2017; this report is updated annually and the report for 2022 has just been published (Cronin, KA, Scott, S, Firth, AU, et al. Annual report to the nation on the status of cancer, part 1: National cancer statistics. Cancer. 2022; 1- 34. https://doi.org/10.1002/cncr.34479). That report indicated between 2014-2018, the incidence of lung cancer declined among males (AAPC=-2.6%) and females (AAPC=-1.1%), and the authors attributed these declines to “the continuous decline in smoking prevalence” (p.28). However, during this same period, the incidence of several “smoking-related cancers” increased among males and/or females: pancreas, kidney and renal pelvis, liver, oral cavity and pharynx, and cervix. Arguably, factors unrelated to smoking are driving the increasing trends in these cancers. Combining these cancers together with lung cancer could distort estimates of the impact of changes in smoking prevalence on cancer rates
Response: We believe it is relevant to analyze the 12 together because these are recognized as the smoking related cancers. We find utility in examining them as a group, though we acknowledge that there is likely to be some heterogeneity among the types. We agree that it would be relevant in future work to explore the contributions of the different types separately and intend to do so.
- The authors are encouraged to revise their tables and figure to be “stand alone” without reference to the text as much as possible. Titles should be descriptive, with details provided in footnotes. In Table 2, it appeared that county population was misplaced in the first column. Careful proof-reading may be needed as the paper is processed by this journal to avoid errors
Response: The population figures error has been corrected. This displacement caused other results to line up incorrectly. These have also been corrected. Table titles have been updated for clarity.
- Had the authors considered limiting their analyses to lung cancer, or to conducting a sensitivity analysis that examined specifically lung cancer? Or perhaps this could be considered as a potential area for future research.
Response: We are indeed exploring further analysis stratified by cancer type with a focus on lung cancer. This work, being a follow up of our prior work which examined the effects of eliminating smoking on the 12 cancers deemed by CDC to be associated with smoking, again focused on the 12 types combined. Additional analyses are planned.
- Because smoking prevalence is a key variable in the analysis, the authors are encouraged to provide further details about their methods for generating estimates of county-level smoking prevalence from BRFSS data. Were there counties for which BRFSS data were not available? If so, how did the methods used to generate county estimates compare with the those used by Berkowitz et al. in 2016 (Cancer Epidemiol Biomarkers Prev (2016) 25 (10): 1402–1410. https://doi.org/10.1158/1055-9965.EPI-16-0244 )?
Response: Dwyer-Lindgren and colleagues provided smoking prevalence estimates for all SEER counties. Their methods of generating small area estimates from BRFSS data were similar to those of Berkowitz and colleagues, however Dwyer-Lindgren provided estimates going back to 1996, while Berkowitz provided results for 2012 only.
Lines 103-10 now read as follows: These smoking prevalence estimates were based on Behavioral Risk Factor Surveillance System (BRFSS) data, which were modeled to generate estimates of county-level smoking prevalence for the US between 1996 and 2012 for ages 18 and over. Estimates for counties that had limited data for a given year were derived via spatial and temporal smoothing techniques which included county and state-level covariates. The variable was defined as “prevalence of current daily cigarette smoking”. Smoking prevalence estimates for 1996 were used, as they allowed for a lag of 20 years for the analysis of the 2016 SEER cancer incidence data.
- In the discussion, the authors state that it is well accepted that air pollution causes some of the same types of cancer as tobacco (line 292-293). The cited references support a causal association between outdoor air pollution and lung cancer but not between outdoor air pollution and the 11 other “smoking-related” cancers. The authors are encouraged to include additional citations, or identify areas for future research, related to air pollution and cancers other than lung cancer.
Response: Thank you for catching this omission. We have added reference to the President’s Cancer Panel report on Reducing Environmental Cancer Risk, which contains a comprehensive list of environmental carcinogens, including air toxics, and associated cancer sites.
- A presentation by Charles Swanton at European Society for Medical Oncology (ESMO) Congress 2022 in Paris generated a great deal of recent media attention because it proposed a novel biologic mechanism by which air pollution could synergistically contribute to cancer development. For example, see this ASCO Post: https://ascopost.com/issues/october-10-2022/mechanism-linking-air-pollution-to-lung-cancer-identified/.
https://oncologypro.esmo.org/meeting-resources/esmo-congress/mechanism-of-action-and-an-actionable-inflammatory-axis-for-air-pollution-induced-non-small-cell-lung-cancer-towards-molecular-cancer-prevention
This work might be relevant to the authors’ discussion of their results and the potential importance of pollution control. A preprint of that study: Swanton, Charles, William Hill, Emilia Lim, Claudia Lee, Clare Weeden, Marcellus Augustine, Kezhong Chen et al. "Non-Small-Cell Lung Cancer Promotion by Air Pollutants." (2022) in available online at: https://assets.researchsquare.com/files/rs-1770054/v1/05c2bee9
Response: We agree that this is a highly relevant finding and we have added the citation and a sentence about it in the Discussion: For example, very recent work by Swanton and colleagues has identified a potentially powerful mechanism through which PM2.5 can contribute to non-small cell lung cancer (NSCLC) in non-smokers, a disease with a high frequency of EGFR mutations (EGFRm). The authors found that PM promotes precursor lung epithelial cells initiated by EGFRm.
Reviewer 2 Report
General comments: This manuscript describes a follow-up to two previous papers describing approaches to evaluate the impact of smoking elimination on the incidence of smoking-related cancers at the county level in the USA. In the current paper, the authors take the next natural step in trying to understand factors that may be predictive of the residual (i.e., non-smoking-related) cancer burden across the set of urbanized SEER counties. The methods are generally sound, and the conclusions appropriate. I have outlined several concerns or suggestions in the Specific Comments below. Overall, I believe this is an impactful paper that will make an important contribution to the literature on environmental carcinogenicity.
Specific comments:
1. An unmentioned limitation of the approach is that county-level smoking data consisted only of prevalence and not intensity or duration. These would be expected to be variable across the counties and could explain some of the residual cancer risk. Please add to limitations section, unless there is a strong rationale not to do so.
2. Line 149: One moderate concern I have was the decision not to include non-urbanized counties in the analysis. According to the US EPA’s latest EQI data, there is also substantial variation in the EQI across counties stratified on urbanization (https://www.epa.gov/sites/default/files/2021-02/eqi_rcc_county_final2.png ). It would be very interesting to have also seen, perhaps in a stratified analysis, whether factors explaining residual smoking-related cancer burden were similar across urbanized and less-urbanized counties.
3. Another concern is that, among the list of 12 smoking-related cancers, there is (according to the IARC Monographs, at least) sufficient or limited evidence of air pollution’s carcinogenicity for only cancers of lung and bladder. The conclusions of the paper about the important role of air pollution would be strengthened if at least the final model results in Table 3 had been stratified to show the associations for lung + bladder vs. the other 10 smoking-related cancers, as one would expect to see a greater impact of these 2 cancers than the other 10 that have not (yet) been demonstrated to be associated with air pollution.
4. Abstract, line 13: after SEER counties, suggest to add “under smoking elimination” to clarify (if indeed this is what is meant).
5. Line 46: urbanized counties tend to also have manufacturing and other industries that may convey exposure to many carcinogens. Suggest to add “or occupational agents” after “air pollution”.
6. Lines 117-121: The decision to conduct a sensitivity analysis comparing the EQI 2006-2010 to the earlier EQI may need some caveats, as these indexes themselves have changed. On its EQI website, the EPA notes: “Because modifications were made to the updated EQI 2006-2010, direct comparisons between EQI 2000-2005 and EQI 2006-2010 should not be made.”
7. line 146: Please explain why county-level smoking prevalence was considered a fixed effect?
8. In Table 2, suggest to indicate which columns represent “low benefit” and “high benefit”, for better concordance with the text.
9. Also in Table 2, Why do the low-benefit (“yes”) counties have a mean percent cancer reduction of >10% from their observed cancer rates? Although clearly lower than the high-benefit (“no”) counties, this mean value of 15.6% seems counter-intuitive, given the information previously provided. Line 218 of the text states that the rate is only 8% lower than the mean for all counties, but since these counties had a higher rate to start with, this seems an inapt comparison. Suggest to carefully consider how this detail is presented throughout the manuscript.
10. In the Discussion, it would have been interesting to see a proposed explanation for the lack of association between PM2.5 and low-benefit county status while the air pollution index was strongly associated with this status.
Author Response
General comments:
This manuscript describes a follow-up to two previous papers describing approaches to evaluate the impact of smoking elimination on the incidence of smoking-related cancers at the county level in the USA. In the current paper, the authors take the next natural step in trying to understand factors that may be predictive of the residual (i.e., non-smoking-related) cancer burden across the set of urbanized SEER counties. The methods are generally sound, and the conclusions appropriate. I have outlined several concerns or suggestions in the Specific Comments below. Overall, I believe this is an impactful paper that will make an important contribution to the literature on environmental carcinogenicity.
Specific comments:
- An unmentioned limitation of the approach is that county-level smoking data consisted only of prevalence and not intensity or duration. These would be expected to be variable across the counties and could explain some of the residual cancer risk. Please add to limitations section, unless there is a strong rationale not to do so.
Response: Thank you for pointing this out. We agree that it should be noted as a limitation. We added to lines 292-3: Smoking data indicated county prevalence; no measure of smoking intensity or duration, which may have varied by county, was available.
- Line 149: One moderate concern I have was the decision not to include non-urbanized counties in the analysis. According to the US EPA’s latest EQI data, there is also substantial variation in the EQI across counties stratified on urbanization (https://www.epa.gov/sites/default/files/2021-02/eqi_rcc_county_final2.png ). It would be very interesting to have also seen, perhaps in a stratified analysis, whether factors explaining residual smoking-related cancer burden were similar across urbanized and less-urbanized counties.
Response: We explored this and after eliminating smoking no rural counties had incidence rates of the 12 smoking-related cancers combined in the top 20th percentile. To clarify this point, we note in the text that all of the low benefit counties were metropolitan.
- Another concern is that, among the list of 12 smoking-related cancers, there is (according to the IARC Monographs, at least) sufficient or limited evidence of air pollution’s carcinogenicity for only cancers of lung and bladder. The conclusions of the paper about the important role of air pollution would be strengthened if at least the final model results in Table 3 had been stratified to show the associations for lung + bladder vs. the other 10 smoking-related cancers, as one would expect to see a greater impact of these 2 cancers than the other 10 that have not (yet) been demonstrated to be associated with air pollution.
Response: Reviewer 1 raised a similar point. As we noted above, we believe there is a good rationale for presenting all 12 smoking-related cancer types as a group because these are all considered “smoking-related cancers”. In future work, we intend to examine cancer types separately.
- Abstract, line 13: after SEER counties, suggest to add “under smoking elimination” to clarify (if indeed this is what is meant).
Response: We have clarified this and the Abstract now reads: County-level cancer rates for 257 metropolitan SEER counties, including the observed results and estimated rates after eliminating smoking, were derived via multilevel regression modeling and age-standardized to the 2016 SEER population.
- Line 46: urbanized counties tend to also have manufacturing and other industries that may convey exposure to many carcinogens. Suggest to add “or occupational agents” after “air pollution”.
Response: Thank you. This change has been made.
- Lines 117-121: The decision to conduct a sensitivity analysis comparing the EQI 2006-2010 to the earlier EQI may need some caveats, as these indexes themselves have changed. On its EQI website, the EPA notes: “Because modifications were made to the updated EQI 2006-2010, direct comparisons between EQI 2000-2005 and EQI 2006-2010 should not be made.”
Response: We added the following on lines 272-75: This reduction may suggest that the longer latency was more appropriate, as expected. However, the Environmental Protection Agency changed the makeup of the EQI measure in the more recent time period and cautions against making direct comparisons between the indices for the two time periods.
- Line 146: Please explain why county-level smoking prevalence was considered a fixed effect?
Response: We added to line 152: Smoking prevalence was modeled as a fixed effect because we assumed the effect of the smoking prevalence would be constant across counties.
- In Table 2, suggest to indicate which columns represent “low benefit” and “high benefit”, for better concordance with the text.
Response: The Table heading has been changed to improve clarity.
- Also in Table 2, Why do the low-benefit (“yes”) counties have a mean percent cancer reduction of >10% from their observed cancer rates? Although clearly lower than the high-benefit (“no”) counties, this mean value of 15.6% seems counter-intuitive, given the information previously provided. Line 218 of the text states that the rate is only 8% lower than the mean for all counties, but since these counties had a higher rate to start with, this seems an inapt comparison. Suggest to carefully consider how this detail is presented throughout the manuscript.
Response: Counties are among those of low benefit if they are in the top 20th percentile of the distribution of post smoking elimination cancer rates. This is not the same as the percent reduction from the county’s actual incidence rate (before smoking was eliminated) compared to its rate after smoking was eliminated. Low benefit counties are counties where incidence remained high in comparison to all other metropolitan counties after smoking was eliminated. Specifically, the low benefit counties are in the top 20% of the distribution of incidence rates post-smoking elimination. We have changed the title of Table 2 to clarify this point.
- In the Discussion, it would have been interesting to see a proposed explanation for the lack of association between PM2.5 and low-benefit county status while the air pollution index was strongly associated with this status.
Response: We agree and have added the following to the Discussion: …the AQI has the advantage of being an integrated measure of many different pollutants, considering not only particulate matter but also many organic and inorganic pollutants including carcinogens. Indeed, the finding that the AQI was more strongly associated with being a low benefit county than PM2.5 suggests that there are other important carcinogens in urban air; in future work we will use more refined measures of air toxics to pursue this hypothesis.